# Universal Detection of Mi^a^ Antigen and Frequencies of Glycophorin Hybrids among Blood Donors in Taiwan by Human Monoclonal Antibodies against Mi^a^ (MNS7), Mur (MNS10), and MUT (MNS35) Antigens

**DOI:** 10.3390/diagnostics11050806

**Published:** 2021-04-29

**Authors:** Meng-Hua Yang, Jen-Wei Chen, Kaito Sayaka, Makoto Uchikawa, Nelson H. Tsuno, Sheng-Tang Wei, Sheng-Mou Hou, Yann-Jang Chen

**Affiliations:** 1Head Office, Taiwan Blood Services Foundation, Taipei 10066, Taiwan; menghua@blood.org.tw (M.-H.Y.); chen@blood.org.tw (J.-W.C.); wei@blood.org.tw (S.-T.W.); shengmou@ms.skh.org.tw (S.-M.H.); 2Institute of Clinical Medicine, National Yang Ming Chiao Tung University, Taipei 11221, Taiwan; 3Institute of Clinical Medicine, National Yang-Ming University, Taipei 11221, Taiwan; 4Laboratory Department, Japanese Red Cross Kanto-Koshinetsu Block Blood Center, Tokyo 135-8639, Japan; S-kaito@ktks.bbc.jrc.or.jp (K.S.); makoto_uchikawa@outlook.jp (M.U.); neltsu26@yahoo.co.jp (N.H.T.); 5Department of Orthopedic Surgery, Shin Kong Wu Ho-Su Memorial Hospital, Taipei 11101, Taiwan; 6Department of Life Sciences and Institute of Genome Sciences, National Yang-Ming Chiao Tung University, Taipei 11221, Taiwan; 7Department of Pediatrics, Taipei Veterans General Hospital, Taipei 11217, Taiwan; 8Department of Education and Research, Taipei City Hospital, Taipei 10341, Taiwan

**Keywords:** human hybridoma, glycophorin hybrids, Mi^a^ antigen, MUT antigen, Mur antigen

## Abstract

Glycophorin hybrids such as GP.Mur are common in Southeast Asians. In Taiwan, clinically significant alloantibodies to the GP.Mur phenotype are the most important issue in blood banks. A large-scale screening of glycophorin hybrids in the Taiwanese population is urgently needed to ensure transfusion safety. Four clones of human hybridomas that secrete anti-Mi^a^, anti-MUT, and anti-Mur were established by fusing human B-lymphocytes and myeloma cells (JMS-3). The specificity of each monoclonal antibody (MoAb) was characterized. Three MoAbs were applied on an Automated Pretransfusion Blood Testing Analyzer (PK7300/PK7400) for donor screening. Genotyping was performed to determine the detailed subgrouping of glycophorin hybrids. Four MoAbs are IgM antibodies. Anti-Mi^a^ (377T) binds to ^46^DXHKRDTYA^54^, ^48^HKRDTYAAHT^57^ peptides, and anti-Mi^a^ (367T) binds to ^43^QTNDXHKRD^51^ peptides (X indicates T, M, or K). Anti-Mur is reactive with ^49^KRDTYPAHTA^58^ peptides. Anti-MUT is reactive with ^47^KHKRDTYA^54^. A total of 78,327 donors were screened using three MoAbs, and 3690 (4.71%) were GP.Mur, 20 (0.025%) were GP.Hut, and 18 (0.022%) were GP.Vw. When the Mi^a^ antigen was introduced as routine screening, the frequency of Mi(a+) among blood donors in Taiwan was 4.66% (67,348/1,444,541). Mi^a^ antigen was implemented as a routine blood testing, and the results were labeled on all red blood cell (RBC) units.

## 1. Introduction

In the MNS blood group system, M and N antigens are located on glycophorin A, and S and s antigens are located on glycophorin B; glycophorins A and B are encoded by the *GYPA* [1] and *GYPB* [2] genes, respectively. These two glycophorins are single-pass sialic acid-rich glycoproteins with numerous *O*-glycans, and GPA also carries an *N*-glycan [3]. Homologous genes *GYPA*, *GYPB*, and *GYPE* share more than a 95% sequence identity, whereas *GYPE* may not encode an RBC membrane component but participates in gene rearrangements resulting in variant alleles [4]. Hybrids of the *GYPA* and *GYPB* genes produce antigenic diversity and new phenotypes. *GYP(A-**ΨB-A)* hybrids encode GP.Vw, GP.Hut., GP.Dane, and so on. Different short portions of pseudo-exon 3 of *GYPB* insert to exon 3 of *GYPA* and result in amino acid changes. *GYP(B-A-B)* hybrids encode GP.Mur, GP.Hop, GP.Bun, GP.HF, and GP.Kip. Exon 3 of *GYPA* inserts to pseudo-exon 3 of *GYPB*, which results in the expression of pseudo-exon 3. Hybrid genes express low-prevalence antigens, such as Mi^a^, Vw, Mur, Hut, Hil, MINY, and MUT.

In Taiwan, the occurrence rate [5] of GP.Mur is high (21.2–88.4%) in several indigenous tribes and 7.3% [6] in the general population. The frequencies are higher in Southeast Asians than in other ethnic groups (9.6% in Thai [7], 6.28% in Hong Kong Chinese [8], 4.7% in American Chinese [9], and 9.5% in Chinese donors [10]). In other ethnic groups, the frequencies of Mi(a+) are 0.22% in Australians [11] and 0.01% in Japanese [12].

Since the first alloantibodies against GP.Mur RBC were reported in Taiwan in 1987 [13], they proved the most prevalent collative alloantibodies in patients [6,14,15,16] and in a donor population [16]. These antibodies are a combination of several specificities or monospecific specificity, such as anti-Vw + Hut + Mur + Mut + Hil or anti-Mur only [6,17]. Alloantibodies, such as anti-Mi^a^, anti-Mur, anti-Hut, and anti-Vw, cause hemolytic disease of the fetus and newborn and delayed hemolytic transfusion reactions [18,19,20,21]. Patients with these antibodies need to be transfused with RBC negative for the corresponding antigens [14]. In previous studies on glycophorin hybrids in Taiwan, monoclonal antibodies (MoAbs) provided from other laboratories and human plasma containing alloantibodies against GP.Mur were applied on small numbers of screening in volunteers, and molecular methods were included when analyzing the sequences of hybrid genes [21,22,23,24]. Confirming glycophorin hybrids by genotyping every donor is impracticable. Several MoAbs have been published, such as anti-Mi^a^, anti-Mur, anti-Vw, and anti-MUT [25,26,27]. Recently, anti-Mi^a^ (GAMA210) has been utilized in donor identifications [11,28]. A combination of serologic typing and genotyping methods is suggested in identifying hybrid glycophorins in donor screenings [10,11,23,28].

The Taiwanese population has a relatively high frequency of alloantibodies against GP.Mur and a higher frequency of the GP.Mur phenotype. An effective assay to identify antigen-negative RBC components is in demand. A comprehensive screening method of glycophorin hybrids among blood donors is also needed. 

In the present study, we established human hybridomas that produce anti-Mi^a^, anti-Mur, and anti-MUT MoAb [29]. MoAbs were characterized using serological assays and a epitope mapping assay. MoAbs were applied on a large-scale screening. This study aims to screen and study the variety of glycophorin hybrids among blood donors in Taiwan.

## 2. Materials and Methods

### 2.1. Hybridoma Cell Lines and MoAb Production

Taiwanese blood donors detected as having alloantibodies against GP.Mur RBCs were identified during a routine antibody screening, and their whole bloods were obtained. Peripheral B lymphocytes obtained from donors with alloantibodies against glycophorin hybrids were transformed by the Epstein–Barr virus and then fused with myeloma cell line JMS-3. Hybridomas were selected by an ouabain-containing HAT medium. Single clones were isolated by limiting dilution [30]. Antibodies secreted by hybridomas were detected by a hemagglutination assay with known GP.Mur RBCs. A culture supernatant (50 μL) was added with 25 μL of RBC suspension in 96-well U-bottom plates, and hemagglutination results were determined after 1 h incubation at room temperature. 

### 2.2. Specificity and Titration Testing of Monoclonal Antibodies

The specificities of three selected IgM monoclonal antibodies were identified using stored panel cells of the phenotypes GP.Vw (Mi I), GP.Hut (Mi II), GP.Mur (Mi III), GP.Hil (Mi V), GP.Bun (Mi VI), and GP.HF (Mi X) as previously described [29].

### 2.3. MoAbs Isotyping

In-house isotyping 96-well microplates were coated with anti-human heavy chain (IgG subclasses 1, 2a, 2b, 3, and IgA and IgM classes) and an anti-human light chain (kappa and lambda) for the qualitative isotype determination of MoAbs. MoAb samples applied to the wells were isotyped by a standard enzyme-linked immunosorbent assay (ELISA). HPR-conjugated anti-human Ig was added as secondary antibody, and the plates were incubated for 1 h at room temperature. After washing, each well was added with TMB substrate for 10 min and then with stop solution. The plates were measured by visual inspection.

### 2.4. Epitope Mapping

Overlapping octapeptides of glycophorin hybrids were synthesized and used for detecting the binding ability of MoAbs from cell culture supernatants. In brief, peptides were reconstituted with DMSO, and a 10 mg/mL stock of each peptide was prepared. Microplate wells were coated (18 h, 4 °C) with 20 μg/mL of each peptide in phosphate-buffered saline (pH 7.4, and coating for 50 μL/well at 4 °C for overnight) and then blocked with 3% skim milk and 300 μL/well at 37 °C for 1 h. Culture supernatant containing MoAbs (50 μL) was loaded and incubated for 1 h at 37 °C. After removing the unbounded MoAbs, HRP-conjugated human IgM (1:5000) (Leadgene, Tainan, Taiwan) was added (1 h, 37 °C), and the reaction was visualized by adding 50 μL of a chromogenic substrate (TMB) at 37 °C for 10 min. The reaction was stopped by adding 50 μL of H_2_SO_4_ to each well. Absorbance was determined at 450 nm using an ELISA plate reader. 

### 2.5. Pilot Study of Serological Testing with Three MoAbs

Anti-Mi^a^ (377T), anti-MUT(366T), and anti-Mur(362T) MoAbs were applied on an Automated Pretransfusion Blood Testing Analyzer, the PK7300/PK7400 Automated Microplate System (Beckman Coulter, Shizuoka, Japan) under the same conditions as previously described [31]. A total of 78,327 donors were screened using anti-Mi^a^, anti-MUT, and anti-Mur. The further workflow of glycophorin hybrid classification is shown in Figure 1.

### 2.6. Genotyping of Glycophorin Hybrids 

The genotype of hybrid alleles was further analyzed in serological glycophorin hybrid samples. Genomic DNA from glycophorin hybrid samples was extracted and subjected to hybrid allele analysis. The genotyping protocol of the *GYP(B-A-B)* hybrid has been previously described [32]. For rare hybrid alleles such as *GYP* HF* and *GYP*Bun*, synthetic gene fragments (gBlocks) were designed to validate this assay. For the ambiguous results, a further sequence analysis was needed. Exons 3 and 4 of hybrid alleles were amplified by primer sets (E3f: ACGTTGGATGGTGCCCTTTCTC-AACTTCTC; E3r: ACGTTGGATGCAGTTAATAGTTGTGGGTGC and E4f: ACGTTGGATGTAAAATGGAATGACTTTT; E4r: ACGTTGGATGGATTTTTTTCTTTGCACATG). Amplicons were examined by Sanger sequencing in both directions performed on an ABI 3730xl DNA analyzer (Applied Biosystems Inc., Foster City, CA, USA).

### 2.7. Mi^a^ Antigen Testing for All Donors

Anti-Mi^a^ (377T) was assayed on the PK7300/PK7400 Automated Microplate System under the same conditions, and MoAb anti-Mi^a^ (377T) was prepared as previously described [31]. The Mi^a^ antigen was introduced as routine donor blood grouping including ABO and RhD.

## 3. Results

### 3.1. Establishment of Cell Lines and Specificities of MoAbs

Four hybridoma cell lines secreting anti-Mi^a^ (377T, 367T), anti-MUT (366T), and anti-Mur (362T) were established [29]. The four were all IgM antibodies. The specificities of MoAbs were identified using GP.Vw, GP.Hut, GP.Mur, GP.Hil, GP.Bun, and GP.HF RBCs provided by the Japanese Red Cross (Table 1). Using undiluted MoAbs from the cell culture supernatant, all four MoAbs showed strong reactions with specific types of glycophorin hybrid RBCs. The titers were tested by a saline phase hemagglutination assay with a serial dilution of MoAbs using GP.Mur RBCs. Titers for each MoAbs ranged from 1:512 to 1:32,768 (Table 1).

### 3.2. Epitope Mapping

Overlapping peptide segments were synthesized from ^40^I to ^55^A of GPA glycophorins and ^40^I to ^59^N of glycophorin hybrids. Anti- Mi^a^ (377T) bound strongly to ^46^DXHKRDTY^53^, ^47^XHKRDTYA^54^, and ^48^HKRDTYAA^55^ peptides across A2 to A3 of GPA and glycophorin hybrids, where X indicates ^47^T, ^47^M, or ^47^K. Anti- Mi^a^ (377T) also bound to 49KRDTYPAH56 and 50RDTYPAHT57 peptides of BΨ3 (Figure 2A). Anti-Mi^a^ (367T) bound to ^43^QTNDXHKR^50^ and ^44^TNDXHKRD^51^ peptides (Figure 2B), which are similar to GAMA210 [25]. However, Anti-Mi^a^ (367T) had no significant difference in binding ability to the ^47^T, ^47^M, and ^47^K of the peptide segments. Anti-MUT (366T) bound strongly to the ^47^KHKRDTYA^54^ of glycophorin hybrids (Figure 3A). Anti-Mur (362T) had equal binding ability to ^49^KRDTYPAH^56^, ^50^RDTYPAHT^57^, and ^51^DTYPAHTA^58^ peptides in the BΨ3 region (Figure 3B).

### 3.3. Screening Test with Three MoAbs

We screened 78,327 donors using anti-Mi^a^ (377T), anti-MUT (366T), and anti-Mur (362T) on PK7300/7400. Of these donors, 3690 (4.81%) were positive to three antigens, which were possibly the GP.Mur, GP.Hop, GP.Bun, and GP.Kip phenotypes. Twenty (0.025%) donors were Mi(a+), MUT-positive, and Mur-negative, which indicated the GP.Hut and GP.HF phenotypes. In addition, 18 donors were only Mi(a+), which were GP.Vw (Table 2). Mi(a+) is the most frequent of these three antigens. Meanwhile, no Mi(a−), MUT-positive, or Mur-positive donors were found in this screening test, such as GP.Dane. Extended phenotyping was only performed on limited samples. Among the donors of each subgroup, 20 donors were confirmed GP.Mur, two donors were confirmed GP.Hut, and three donors were confirmed GP.Vw by an additional assay with anti-Hil, anti-NEV, and anti-Vw antibodies (Gifts from Dr. Uchikawa, JRC). The three MoAbs we produced showed stable hemagglutination results on PK7300/7400 and were suitable for routine donor screening.

### 3.4. MALDI-TOF Genotyping for Glycophorin

We selected 676 donors from the screening test (Figure 1) for MALDI-TOF genotyping. Only seven samples showed ambiguous results on MALDI-TOF genotyping. Genotypes of these samples were analyzed by Sanger sequencing. Six hundred and thirty-eight donors who were all positive for Mi^a^, MUT, and Mur antigens had *GYP*Mur* alleles. Among them, 605 (94.8%) were *GYP*Mur* homozygotes, and 33 (5.2%) were *GYP*Mur* homozygotes. All 20 donors who were positive for the Mi^a^ and MUT antigens were *GYP*Hut* heterozygotes. Eighteen donors who were Mur-positive were *GYP*Vw* heterozygotes. No *GYP*Hop*, *GYP*Bun*, *GYP*Kip*, *GYP*Hut*, or *GYP*HF* alleles were detected.

### 3.5. Population Frequency of Mi^a^ Antigen

On the basis of the screening test, the Mi^a^ antigen is the most prevalent of the three hybrid antigens in our population. Since December 2018, Mi^a^ antigen testing was incorporated in routine ABO and RhD blood group screening using the PK7300/PK7400 Automated Microplate System for all donations in Taiwan. From 2019 to 2020, 67,348 (4.66%) donors were Mi(a+) out of 1,444,541 donors (Figure 4). The distribution of Mi(a+) donors was uneven. The frequencies of Mi(a+) were much higher in two eastern counties, Hualien (18.18%) and Taitung (18.53%), whereas those in other areas ranged from 3.06% to 5.09%. After introducing the Mi^a^ antigen as routine testing, the results were also provided on red cell components. This had decreased 46% of the requests for Mi(a−) red cell components from the blood banks of hospitals.

## 4. Discussion

Providing GP.Mur antigen-negative RBC to blood recipients with this alloantibody decreases the transfusion reaction rates [14]. Considering the lack of antisera suitable for the large-scale screening of glycophorin hybrids among all donors, we conducted this study to establish the four human hybridoma cell lines producing IgM anti-Mi^a^, anti-MUT, and anti-Mur from donors who had alloantibodies against GP.Mur RBCs with the support provided by Japanese Red Cross, Kanto-Koshinetsu Block Blood Center, Tokyo, Japan. The four MoAbs, anti-Mi^a^(377T), anti-Mi^a^(367T), anti-MUT(366T), and anti-Mur(362T), were serologically confirmed in specificity by using GP.Vw, GP.Hut, GP.Mur, GP.Hil, GP.Bun, and GP.HF RBCs.

The hypotheses of epitopes for glycophorin hybrid-related antibodies were published in 1992 [33]. MoAb anti-Mi^a^ secreting hybridoma cell lines from human B-lymphocytes were the first identified. Our data suggested that two groups of anti-Mi^a^ were present in different donors. Anti-Mi^a^ (377T) recognized three regions of peptide sequences, including A3 of GPA: ^45^DTHKRDTYAA^55^; hybrid BΨ3-A3 of GP(A-B-A): ^46^DMHKRDTYAA^55^ and ^46^DKHKRDTYAA^55^; and BΨ3 of GP(B-A-B): ^49^KRDTYPAHT^57^. Meanwhile, anti-Mi^a^ (367T) recognized peptides across A2-A3 of GPA: ^43^QTNDTHKRD^53^, A2-BΨ3-A3 of GP(A-B-A): ^43^QTNDMHKRD^51^, and B2-BΨ3 of GP(B-A-B): ^43^QTNDKHKRD^51^. Anti-Mi^a^ (367T) had similar epitopes to the murine anti-Mi^a^ (GAMA210) published in 2001 [25], which were ^44^TNDKHKRD^51^ and ^43^QTNDMHKR^50^. Epitopes of anti-Mi^a^ (377T) and anti-Mi^a^ (367T) corresponded to the possible sequences of Mi^a^ antigen ^43^QTND(M/K)HKRDTY^53^ [4,33]. Notably, anti-Mi^a^ (377T) and anti-Mi^a^ (367T) also recognized the synthetic peptides of GPA but did not agglutinate RBCs with normal GPA. Similarly to GAMA210 [25], the N-glycan linked to Asn45 on the RBC surface might mask or alter the epitope detected by anti-Mi^a^ (377T) and anti-Mi^a^ (367T). Anti-MUT (366T) bound strongly to ^47^KHKRDTY^53^ of GP(A-B-A): GP.Hut and GP(B-A-B): GP.Mur, GP.Hop, GP.Bun, GP.HF, and GP.Kip. The murine anti-MUT(CBC-412) published by Uchiwaka et al. [27] bound to peptides ranging ^44^TNDKHKRDTY^53^ (personal communication). Donors can produce naturally occurring monospecific anti-Mur only [17]. Human MoAb anti-Mur(HIRO-138) [26] and murine anti-Mur (CBC-431) [27] were published in 2000 and 2015 by Uchikawa et al., respectively. The anti-Mur (362T) we produced recognized ^49^KRDTYPAHTA^58^, which contained a part of the region of ^53^YPAHTANE^60^ of GP(B-A-B): GP.Mur, GP.Hop, GP.Bun, and GP(A-B-A): GP.Dane [4,33]. Several possibilities may affect the results between the epitope prediction from the binding of synthetic peptides and from agglutinations of RBCs. MoAbs may react with different “facets” of the same antigenic determinant [34]. Glycosylation on adjacent amino acid residues may also affect the reactivity of MoAbs [35].

In the present study, three IgM MoAbs, anti-Mi^a^ (377T), anti-MUT (366T), and anti-Mur (362T), were applied on PK7300/PK7400 at the same time for screening 78,327 random donors. A detailed subgrouping of glycophorin hybrids was performed by genotyping on a part of selected donors. Among them, 4.71% were GP.Mur, 0.025% were GP.Hut, and 0.022% were GP.Vw. Although the frequency of GP.Mur was lower than the previous report of 7.3% [6] and 5.6% [24] in Taiwan, the most frequent were glycophorin hybrids. The subgroups of glycophorin hybrids discovered in our population were also various. The GP.Vw and GP.Hut we detected were not reported in publications of Taiwanese origin [5,22,23,24]. The GP.Hil and GP.Hop [22] published before were not detected this time, whereas GP.Hil can only be detected by anti-Hil and anti-MINY, which were not included in our study. Similarly, the study of Asian American type O donors who were initially screened with anti-Mi^a^ (GAMA210) and subjected to genotyping reported GP.Mur, GP.Bun, GP.Vw, and GP.Hut [28]. After screening with an anti-Mur and MLPA genotyping analysis of 528 donors in a Chinese population, 9.5% GP.Mur donors and 0.19% GP.Bun donors were identified [10]. In recent studies among Japanese donors [12], MoAb anti-Mi^a^(CBC-172) and polyclonal antiserum anti-Hil were used to screen and classify detailed MNS glycophorin hybrids. The full panel of typing sera listed in previous studies [11,12] was not easy to access. In comparison with our study, it used anti-Hop, anti-Hil, anti-Bun, and anti-Anek antisera to serologically classify detailed subgroups of the 3690 samples with the Mi^a^, MUT, and Mur antigens. Therefore, applying a combination of serologic typing and genotyping methods for identifying glycophorin hybrids in donor screenings is effective and suitable. The MALDI-TOF MS-based method we applied identified seven genotypes of glycophorin hybrids [32]. In this assay, we also designed gblock fragments of *GYP* HF* and *GYP*Bun* combined with human DNA of *GYP* Mur*, *GYP* Hut*, and *GYP* Vw* and common *GYPA* and *GYPB* as controls in each genotyping assay. After genotyping 638 donors randomly selected out of 3690 donors (Figure 1), only *GYP* Mur* was identified. To identify possible cases of other glycophorin hybrids, further study should be performed to increase the number of individuals under genotyping analysis, especially for Group O donors. RBCs from the full classification of glycophorin hybrids of group O donors are useful as reagent red cells for identification of alloantibodies against glycophorin hybrids [17].

Studies of glycophorin hybrids in Taiwan [5,22,23,24], including this paper, showed that GP.Mur is the most prevalent subgroup and that the Mi^a^ antigen can be detected in most glycophorin hybrids. Therefore, we incorporated the Mi^a^ antigen as routine blood group screening for all donations in Taiwan using anti-Mi^a^ (377T). From 2019 to 2020, the frequency of Mi(a+) among blood donors was 4.66% (67,348/1,444,541). The analysis of the demographic distribution of Mi(a+) donors revealed that the frequencies of the eastern counties Hualian and Taitung are 18.18% and 18.53%, respectively. Our data corresponded to the fact that the high frequencies of GP.Mur in several Taiwanese aboriginal tribes are distributed on eastern Taiwan [5].

## 5. Conclusions

In summary, we established four human hybridomas secreting IgM anti-Mi^a^ (377T, 367T), anti-MUT (366T), and anti-Mur (362T). The epitopes of MoAb were confirmed. This result proved the isolations of anti-Mi^a^-secreting B-lymphocytes from humans. Serological screening followed by glycophorin hybrid genotyping identified GP.Vw (0.022%), GP.Hut (0.025%), and GP.Mur (4.71%). Screening all donations in 2019–2020 revealed a Mi(a+) frequency of 4.66% among blood donors in Taiwan, and the Mi^a^ antigen results were provided on RBC components. The selection of Mi(a−) RBC components is now easier for pretransfusion compatibility tests at hospital blood banks. It has a high incidence of alloantibodies against GP.Mur in the Taiwanese population [16], and this approach improves the transfusion safety and lowers the transfusion reaction rate [14].

## 6. Patents

Taiwan Patent I698643 “Antibody and antibody fragments, kit and method for detecting Miltenberger blood group antigen”.

## Figures and Tables

**Figure 1 diagnostics-11-00806-f001:**
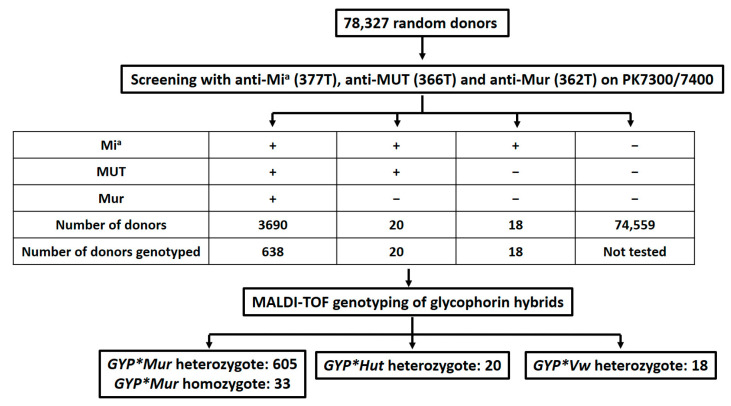
Summary of screening workflow.

**Figure 2 diagnostics-11-00806-f002:**
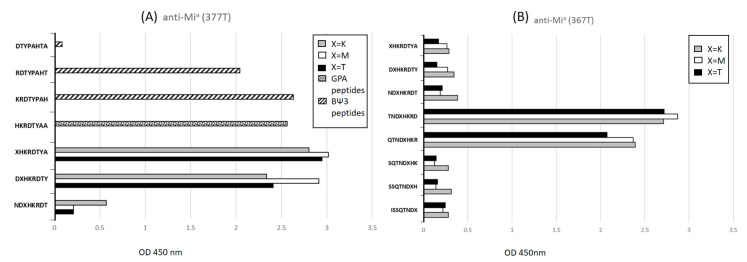
Binding of (**A**) anti-Mi^a^(377T) and (**B**) anti-Mi^a^ (367T) to overlapping peptide segments synthesized from ^40^I to ^55^A of GPA glycophorins, and ^40^I to ^58^A of glycophorin hybrids.

**Figure 3 diagnostics-11-00806-f003:**
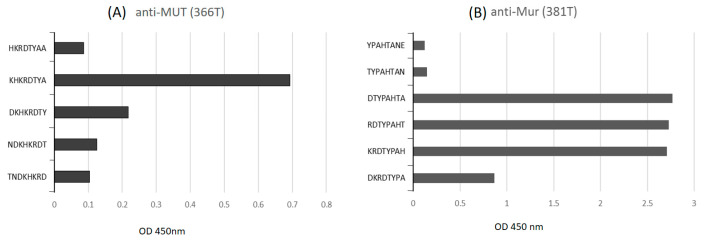
Binding of (**A**) anti-MUT(366T) and (**B**) anti-Mur (381T) to overlapping peptide segments synthesized from ^40^I to ^55^A of GPA glycophorins, and ^40^I to ^59^N of glycophorin hybrids.

**Figure 4 diagnostics-11-00806-f004:**
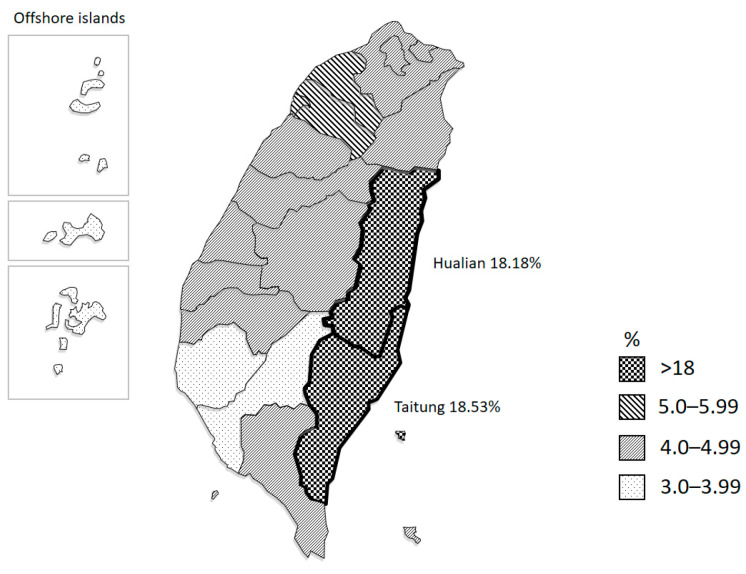
Geographical distribution of the proportion of Mi(a+) donors in Taiwan. The proportions of Mi(a+) donors are much higher in eastern Taiwan.

**Table 1 diagnostics-11-00806-t001:** Identification and isotype of MoAbs.

RBCs	anti-Mi^a^ (377T)	anti-Mi^a^ (367T)	anti-MUT (366T)	anti-Mur (381T)
GP.Vw	4+	4+	0	0
GP.Hut	4+	4+	4+	0
GP.Mur	4+	4+	4+	4+
GP.Hil	0	0	0	0
GP.Bun	4+	4+	4+	4+
GP.HF	4+	4+	4+	0
Titer ^1^	1:2048	1:2048	1:512	1:32,768
Isotype	IgM	IgM	IgM	IgM

^1^ Hemagglutination titration using GP.Mur RBCs.

**Table 2 diagnostics-11-00806-t002:** Pilot study of serological testing with three MoAbs on 78,327 donors.

Antigens of Glycophorin Hybrids			Possible Phenotypes	Number of Donors	Frequency (%)
Mi^a^	MUT	Mur			
+	+	+	GP.Mur, GP.Hop, GP.Bun, GP.Kip	3690	4.71
+	+	−	GP.Hut, GP.HF	20	0.025
+	−	−	GP.Vw	18	0.022
−	−	−		74,599	95.2

## Data Availability

Raw data are available from the first author on reasonable request.

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
