# Peer review of "Universal Detection of Mia Antigen and Frequencies of Glycophorin Hybrids among Blood Donors in Taiwan by Human Monoclonal Antibodies against Mia (MNS7), Mur (MNS10), and MUT (MNS35) Antigens"

_diagnostics, 2021, doi:10.3390/diagnostics11050806_

Round 1
Reviewer 1 Report
In this paper the authors carry out a genotype analysis of over the blood of 78,000 Taiwanese donors to determine the distribution of the Mia antigen.
My expertise is outside the field of haematology (in biostatistics) so I will restrict my comments to the presentation of the results.
The sample size seems adequate for the descriptive nature of the paper.
In Figure 3, the units of the horizontal axis should be given. There is also an extra full stop at the end of the figure caption.
In Figure 4, the caption seems to lack detail. Should it read “Geographical distribution of the proportion of Mia donors in Taiwan. The proportions of Mia donors is much higher in eastern Taiwan.”
Author Response
Thank you for your kind letter of “diagnostics-1190728”. We revised the manuscript in accordance with the reviewers’ comments, and carefully proof-read the manuscript.
Figure 3:
Thanks for the comment. We have changed the “Absorbance” to “OD 450 nm”in the horizontal axis of figure 3. This term is more comprehensive. Optical density (OD) is the measure of absorbance, and is defined as the ratio of the intensity of light falling upon a material and the intensity transmitted. Absorbance readings are calculated from a ratio of the intensity of light transmitted through the sample (I) to the intensity of light transmitted through a blank (Io). This ratio results in a unitless value. Absorbance = log (Io/I)
The extra stop is also deleted.
Figure 4:
Thanks for the comment. Figure legends are revised as suggestion.
Reviewer 2 Report
In this paper, JW Chen et al focus on the detection of Mia antigen and frequency of glycophorin hybrids among blood donors in Taiwan. The study is interesting and well organized.
Author Response
We thank the Reviewer’s positive comment.